# Phosphate Dysregulation and Metabolic Syndrome

**DOI:** 10.3390/nu14214477

**Published:** 2022-10-25

**Authors:** Nikolay Mironov, Mainul Haque, Azeddine Atfi, Mohammed S. Razzaque

**Affiliations:** 1Department of Pathology, Lake Erie College of Osteopathic Medicine, Erie, PA 16509, USA; 2Unit of Pharmacology, Faculty of Medicine and Defense Health, National Defense University of Malaysia, Kuala Lumpur 57000, Malaysia; 3Department of Pathology, Virginia Commonwealth University, Richmond, VA 23298, USA

**Keywords:** phosphate burden, obesity, hypertension, diabetes, insulin

## Abstract

Phosphorus is one of the most abundant minerals in the human body. It is essential for almost all biochemical activities through ATP formation, intracellular signal transduction, cell membrane formation, bone mineralization, DNA and RNA synthesis, and inflammation modulation through various inflammatory cytokines. Phosphorus levels must be optimally regulated, as any deviations may lead to substantial derangements in glucose homeostasis. Clinical studies have reported that hyperphosphatemia can increase an individual’s risk of developing metabolic syndrome. High phosphate burden has been shown to impair glucose metabolism by impairing pancreatic insulin secretion and increasing the risk of cardiometabolic disorders. Phosphate toxicity deserves more attention as metabolic syndrome is being seen more frequently worldwide and should be investigated further to determine the underlying mechanism of how phosphate burden may increase the cardiometabolic risk in the general population.

## 1. Introduction

If untreated, long-standing obesity, hypertension, and altered glucose metabolism usually progress to metabolic syndrome. The criteria provided by the National Cholesterol Education Program Adult Treatment Panel III (NCEP ATP III) has been commonly used to diagnose metabolic syndrome. It states that a metabolic syndrome is a group of conditions that occur together and increase the risk of type 2 diabetes mellitus, heart disease, and stroke [1]. Metabolic dysfunction becomes metabolic syndrome when any 3 of the 5 criteria are met which include: (1) blood pressure values of systolic ≥130 mmHg and/or diastolic ≥85 mmHg; (2) fasting glucose ≥ 100 mg/dL; (3) elevated triglycerides ≥ 150 mg/dL; (4) HDL <40 mg/dL in men or <50 mg/dL in women and (5) waist circumference > 40 inches for men and 35 inches in women [1]. This article will discuss how phosphate dysregulation can be associated with the criteria mentioned earlier for metabolic syndrome.

Phosphorus (P) is one of the most abundant minerals, along with sodium (Na^+^), calcium (Ca^2+^), magnesium (Mg^2+^), and potassium (K^+^) in humans. Throughout this article, the term ‘phosphate’ (PO_4_^3−^) will be interchangeably used to describe this element. Inorganic phosphate is essential for intracellular signal transduction (phosphorylation), energy metabolism (ATP), cell membrane formation (phospholipids), synthesis of DNA and RNA (nucleic acid), and bone mineralization (hydroxyapatite) [2,3,4]. Since phosphate actively regulates the phosphorylation reactions, in vitro studies suggest that its optimal intracellular concentration is required for the phosphorylation of insulin receptor tyrosine-kinase and subsequent cellular insulin signaling. Although further studies are needed, dysregulation of phosphate balance potentially impairs insulin signaling to trigger insulin resistance with reduced cellular glucose utilization. Nguyen et al. demonstrated in vitro that oxidative stress related to mitochondrial phosphate uptake is responsible for defective insulin synthesis and secretion, as secretion is sensitive to increased ROS induced in mitochondria due to hyperphosphatemia [5].

Phosphate absorption, reabsorption, and resorption occur in the small intestine, kidney, and bone. Coordinated crosstalk among these organs maintains normal homeostatic phosphate balance [6]. Phosphate-regulating hormones, including fibroblast growth factor-23 (FGF23), calcitriol, and parathyroid hormone (PTH), actively regulate phosphate homeostasis [2,3]. Excessive phosphate consumption and/or impaired renal excretion results in a positive phosphate balance. The maximum phosphate load is derived from meat, poultry, fish, hard cheeses, egg yolk, and nuts [7]; when phosphate-based preservatives are added to these commonly consumed foods, the amount of phosphate ingestion markedly increases [8]. For healthy individuals, the adverse consequence of phosphate overload is more likely to develop than phosphate depletion. An epidemiologic study on non-diabetic individuals found that increased calcium and calcium-phosphate product are associated with a higher risk of developing type 2 diabetes independently of measured glucose, insulin secretion or insulin resistance [9], though subsequent analysis found that only elevated serum calcium levels and calcium-phosphate product were associated with developing type 2 diabetes [9]. After adjustment for glucose tolerance and insulin sensitivity/secretion, high phosphate concentration is associated with future diabetes development [9]. In a separate study with 460 normal-weight adults aged 18–35 years, higher levels of calcium and lower levels of phosphate were shown to be significantly associated with metabolic syndrome [10].

## 2. Phosphate Regulation

The recommended dietary allowance of phosphorus for adults and elderly individuals in the United States (U.S.) is 700 mg/day. However, approximately one-third of the U.S. adult population consumes more than the recommended amount [11]. Studies on schoolchildren have shown that increased consumption of phosphate enhances the risk of gingivitis by increasing salivary interleukin (IL)-1β while decreasing IL-4 levels [12]. IL-1β regulates the innate immune process via leukocytic pyrogenicity, inducing various acute-phase response proteins and lymphocyte-activating factors (LAFs) [13]. IL-4 regulates antibody production, hematopoiesis and inflammation, and the development of effector T-cell responses [14]. Moreover, a significant association has been detected between salivary phosphate and C-reactive protein levels in children [15,16,17]. 

FGF23, a master regulator of phosphate homeostasis, is synthesized by the bone cells. Translated FGF23 protein can undergo two post-translation modifications: (1) O-glycosylation with GALNT3 on Threonine 178 and (2) phosphorylation by FAM20C at Serine 180 [18,19]. O-glycosylation prevents intact FGF23 from cleavage, while phosphorylation by the extracellular serine/threonine protein kinase FAM20C can cleave FGF23 into N-terminal and C-terminal fragments in bone cells. Bone-derived FGF23 binds to the FGF receptor (FGFR)/αKlotho complex on renal proximal tubular epithelial cells to suppress sodium-dependent phosphate co-transporter type IIa (NaPi-2a) expression [20,21]. Consequently, the reduced activity of NaPi-2a results in reducing renal phosphate reabsorption and increasing urinary phosphate excretion. FGF23 also reduces renal 1α-hydroxylase expression and increases 24-hydroxylase expression in the proximal tubular epithelial cells to reduce vitamin D activation (Figure 1) [20,21]. Negative feedback control between bone and kidney partly regulates FGF23 production. Reduced phosphate and vitamin D levels exert inhibitory effects on bone-derived FGF23 synthesis by reducing extracellular phosphate signaling and VDR activities on the bone cells. The absence of FGF23 or klotho system leads to uncontrolled hyperphosphatemia and increased risk for systemic dysfunction in metabolic syndrome [5,22].

Dysregulated phosphate homeostasis increases the risk of systemic organ dysfunction in metabolic syndrome. A higher phosphate burden could generate a chronic low-grade inflammation that promotes the cluster of conditions that are commonly encountered in metabolic syndrome, including vascular stiffness (hypertension), altered glucose homeostasis (diabetes), and abnormal lipid metabolism (obesity) [23]. Of importance, even a subclinical phosphate burden could generate proinflammatory cytokines and free radicals to create a low-grade inflammatory microenvironment [23,24]. When adjusting for elevated calorie intake, elevated phosphate intake was directly associated with increased salivary IL-1β concentration [12]. IL-1β has been shown to inhibit pancreatic β cell function via promoting Fas-triggered apoptosis following NF-κB transcription factor activation, leading to impaired insulin secretion [25]. The serum phosphate level does not always reflect the accurate picture of total body phosphate content, as the intravascular pool of phosphate represents less than 1% of total body phosphate. Clinical and subclinical chronic inflammatory responses are the underlying mechanism of various commonly encountered systemic diseases, including obesity, metabolic syndrome, hypertension, and cardiovascular disorders [26,27]. 

The parathyroid gland secretes PTH, and its central role is to balance calcium and phosphate in the blood. It is secreted to raise calcium levels when too low or when phosphate levels are too high [28]. One of the actions of PTH is to increase calcium reabsorption in the proximal convoluted tubule (PCT) of the kidney while simultaneously decreasing phosphate reabsorption in the PCT. PTH also increases the production of 1α-hydroxylase from the kidneys, which catalyzes the reaction of calcifediol into calcitriol (active vitamin D) [28]. The reasoning for lowering phosphate and not just raising plasma calcium concentration is that phosphate ions readily form insoluble salts with calcium, resulting in a decrease in plasma calcium [28]. PTH and its related signaling peptide PTHrP induce glycolysis and fatty acid oxidation in bone cells and drive lipolysis and thermogenic programming in adipocytes [29]. Osadnik et al. identified a significant association between calcium-phosphate and metabolic syndrome in normal-weight individuals (10). They found hypercalcemia and hypophosphatemia in the metabolic syndrome group but not in the group of healthy individuals with normal weight. 

Phosphate toxicity could directly trigger chronic inflammation by inducing cytotoxicity and disrupting subcellular signaling events [30,31,32,33]. Moreover, commonly consumed high phosphate-added processed foods and drinks adversely alter the gut microbial ecosystem to promote the cluster of conditions that are widely encountered in metabolic syndrome [34]. Noteworthy, recent studies have found a possible association between gut microbiota and metabolic syndrome, possibly by promoting inflammation [35,36]. 

Phosphate burden in hypertension, type 2 diabetes mellitus, obesity and metabolic syndrome might not always clinically reflect as hyperphosphatemia. Increased phosphate burden in individuals with normophosphatemia may have pathological consequences, as abnormal phosphate balance could induce vascular stiffness as well as impair glucose and lipid metabolism [37]. In vitro studies have shown that oxidative stress related to mitochondrial phosphate uptake is responsible for defective insulin synthesis and secretion [5]. Tonelli et al. identified a relationship between upper limits of normal serum phosphate and increased risk of cardiovascular events, heart failure, and all-cause death in people with prior myocardial infarctions, most of whom had normal serum phosphate and kidney functions [38]. They noted a significant association between baseline serum phosphate levels and the age, race, and sex-adjusted risk. Individuals with 3.5 mg/dL serum phosphate levels had 1.55 times higher hazard ratios than those with levels below 2.8 mg/dL [38]. Of relevance, a serum level of <4 mg/dL is considered within the accepted normal range of phosphate. According to Mayo Clinic Laboratories, the standard range of serum phosphorus for adults is 2.5–4.5 mg/dL.

## 3. Phosphate Burden and Glucose Metabolism

Glucose-induced insulin secretion by the pancreatic β-cell is accompanied by the significant efflux of intracellular inorganic phosphate (∼50% of the cellular content), commonly known as “phosphate flush” [39]. Of relevance, “phosphate flush” has specificity towards pancreatic β-cells, as other islet cells or the exocrine pancreas are not involved with this phenomenon [40]. Studies have also shown that a higher phosphate burden can reduce pancreatic β-cell viability [41]. The “phosphate flush” in the pancreatic β-cell might be a protective phenomenon to evade cellular phosphate burden [42]. Hyperphosphatemia alters how the human body metabolizes glucose, putting individuals at risk for developing metabolic syndrome [41]. Phosphate enters insulin-releasing cells in the pancreas via sodium-dependent phosphate co-transporters (type II and type III) [5]. Phosphate then enters mitochondria by an intricate process involving various proteins such as phosphate carrier (PiC), dicarboxylate carrier (DIC), and uncoupling protein-2 (UPC2) [5]. High phosphate levels are toxic to various cell types as it increases mitochondrial membrane potential, superoxide generation, activation of caspases and eventual cell death via intrinsic apoptosis pathway [5,41]. Beta islet cells release insulin from the pancreas and have minimal antioxidation protective mechanisms [5,41]. Increases in ROS production activate the unfolded protein response (UPR), which causes phosphorylation of double-stranded RNA-dependent protein kinase-like ER kinase (PERK) and eukaryotic translation initiation factor 2α (eIF2α) due to high phosphate levels, causing attenuation of insulin translation [5,41]. A decrease in insulin release allows high glucose levels to remain in the circulatory system, further exacerbating diabetes [41,43]. Phosphate burden should be controlled actively to attenuate its negative effects on insulin and help prevent one of the ways an individual may develop metabolic syndrome (fasting glucose ≥ 100 mg/dL).

In nondiabetic subjects, hypophosphatemia has been shown to be associated with reduced insulin sensitivity [44]. A close interaction exists between glucose and phosphate metabolism. Earlier studies have shown that insulin-dependent cellular glucose uptake is accompanied by phosphate translocation from the extracellular compartment into the intracellular compartment [45]. Hypophosphatemia could also impair pancreatic insulin secretion [46,47]. Moreover, in an experiment conducted by Defronzo et al., they noticed that when exogenous insulin and glucose were infused at a constant rate, the infusion rate of glucose was 43% lower in the hypophosphatemic group than in the control group even though the insulin clearance rate was comparable in both groups [48]. This indicates that hypophosphatemia-associated impaired glucose metabolism is related to decreased tissue sensitivity to insulin [48]. In a case-control study conducted on elderly type 2 diabetes patients with renal impairment, high serum phosphate contributed to vascular calcifications, arterial stiffness, and cardiovascular mortality. Hyperphosphatemia was also correlated with increased metabolic disturbances, such as significantly higher uric acid, serum glucose, i-PTH, and albuminuria, compared to low serum phosphate levels and metabolic disturbances compared to non-diabetic patients [49].

As stated, the change in dietary habits and an increased reliance on processed foods resulted in phosphate overload in more than one-third of the adult population in the U.S. [11]. Modern Western diets contain a higher amount of artificial phosphate (carbonated beverages, processed meat, processed cheese), which is almost completely absorbed in the intestine, as opposed to natural phosphate-containing foods, which are absorbed only around 40%–60% [50]. The chemical analyses of additive-containing processed meat and fish had a phosphate level of around 14.4 mg per gram of protein, while unprocessed foods lacking additives had a phosphate level of around 9.1 mg per gram of protein [51,52]. Also, the absorption of meat-derived phosphate is relatively higher than that of phosphate from plant sources [53]. Analysis of the 2007–2014 U.S. National and Health Nutrition Examination Survey (NHANES) found a positive correlation between BMI with the amount of phosphate and sodium consumption [54]. 

## 4. Phosphate Burden and Cardiometabolic Disorders

Prolonged dietary phosphate overload could induce subclinical inflammatory responses to promote vascular stiffness, insulin resistance, and obesity, likely enhancing the likelihood of developing type 2 diabetes and cardiometabolic disorders [55,56] (Figure 2). In ApoE knockout mice, high dietary phosphate intake accelerates atherogenesis independent of vascular calcification [57]. Moreover, high phosphate consumption impairs endothelial cell functions in young, healthy men [58]. In vitro studies have shown that a high phosphate microenvironment could impair endothelial cell survival, migration, capillary tube formation, and angiogenesis [59]. In the Cholesterol and Recurrent Events (CARE) study, patients with serum phosphate >3.5 mg/dL had a higher adjusted hazard ratio for death by myocardial infarction than those with serum phosphate <3.5 mg/dL [38]. A study conducted on 1989 Chinese patients with ST-segment elevation myocardial infarction (STEMI) showed a high serum phosphate level was associated with all-cause mortality among STEMI patients with preserved renal function [60]. According to Huang et al., hyperphosphatemia had a greater influence on increased systolic blood pressure, diastolic blood pressure, and pulse pressure in patients undergoing hemodialysis when adjusting for potential confounding variables such as anti-hypertensive medication use, demographics, medical history, and dialysis non-adherence [61]. The likely mechanism is hardened arterial walls caused by mineral deposition and vessel non-distensibility [61]. This vessel noncompliance is one-way hyperphosphatemia that causes individuals to meet the blood pressure criteria of metabolic syndrome with blood pressure values of systolic ≥130 mmHg and/or diastolic ≥85 mmHg.

Although low or high phosphate balance may be associated with altered glucose metabolism to induce metabolic syndromes, excessive phosphate-containing food consumption in healthy individuals is more of a health risk than deficiency. For instance, in a study conducted on 92,756 individuals with normal kidney function, higher serum phosphorus levels influenced all-cause mortality of the studied cohorts [62]. Similarly, hospitalized patients with hyperphosphatemia at admission were associated with an increased risk of in-hospital mortality [63].

Human and experimental studies have consistently demonstrated a higher phosphate burden induces cardiovascular calcification, hypertension, musculoskeletal disorders, and premature aging [4]. In a study conducted on 104 diabetic patients, the serum phosphate level was markedly higher in cardiovascular disease patients [43]. Simple regression analyses found a positive correlation between serum levels of phosphate and serum levels of total cholesterol (*p* < 0.05, r = 0.35), triglyceride (*p* < 0.05, r = 0.75) and fasting glucose (*p* < 0.05, r = 0.75) [43]. According to Mahmud et al., all cardiac patients in their study had poor glycemic control and found a linear and significant increase in triglycerides in these patients [43]. As previously mentioned, hyperphosphatemia decreases insulin secretion leading to hyperglycemic states. In a similar line of experimental study, genetically inducing phosphate toxicity in klotho deficient mice showed significantly reduced pancreatic insulin content as compared with wild-type mice [64]. Inadequate insulin secretion is also noted in hyperphosphatemic patients with chronic kidney disease. Retrospectively analyzing 48 diabetic patients, hyperphosphatemia was consistently detected during diabetic ketoacidosis episodes; the serum phosphate level in these patients also positively correlated with serum glucose level [65]. A serum phosphate level of 3.5 mg/dL and above showed an association with a serum triglyceride level of >150 mg/dL, satisfying yet another criterion for metabolic syndrome [43]. In a study conducted on 1408 females and 1096 males, compared to non-smoking subjects, smoking males and females with diabetes patients had higher serum phosphate and triglyceride levels [66]. Such association between smoking and serum phosphate and triglyceride levels existed even after adjusting for age and cardiovascular risk factors through multiple linear regression analyses [66]. In a follow-up study by the same group, however, they reported that low serum phosphate level was also associated with all-cause mortality in smokers [67]. A clinical association between phosphate burden and impaired insulin secretion has been widely documented, and further experimental studies will be needed to clarify underlying molecular mechanisms. 

## 5. Conclusions

Mineral ion deficiency in clinical medicine, including Ca^2+^, Mg^2+^, zinc (Zn^2+^), and iron (Fe^3+^), is more common than generally acknowledged in elderly individuals and disease-related malnutrition in hospitalized patients [68,69,70,71,72,73]. However, individuals with a higher phosphate burden are at a greater risk of developing complications associated with metabolic syndrome. Since the global prevalence of hypertension, obesity, and type 2 diabetes is alarmingly increasing, and dependency on phosphate-rich processed foods is growing, reducing the phosphate burden by adopting a healthy diet and lifestyle may be a sustainable and inexpensive intervention to reduce the disease burdens associated with metabolic syndrome. Moreover, a lack of awareness of artificially added phosphate-rich food and drinks makes it harder for consumers to monitor phosphate uptake [74,75]. The absence of the amount of phosphate in the nutritional ingredient lists also contributes to dietary phosphate overload, even among health-conscious individuals. Hyperphosphatemia has been consistently detected in the advanced stage of both human and experimental models of diabetes mellitus [76,77].

Future prospective studies to reduce the phosphate burden before and during the appearance of the cluster of conditions associated with metabolic syndrome are required to determine the potential adverse effects of phosphate burden on cardiometabolic risk in the general population. 

## Figures and Tables

**Figure 1 nutrients-14-04477-f001:**
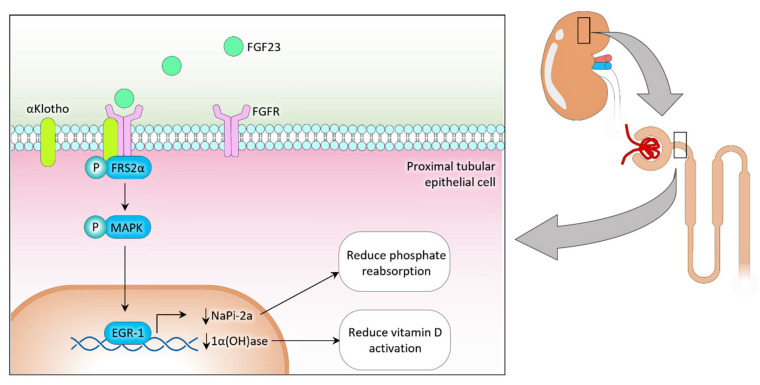
Osteoblasts and osteocytes mainly produce FGF23, which binds to the αKlotho/FGFR complex to activate downstream signaling events and suppress the expression of NaPi-2a and 1a(OH)ase to inhibit phosphate reabsorption and reduce vitamin D activation processes, respectively, in the proximal tubular epithelial cells. FRS2α: fibroblast growth factor receptor substrate 2α; MAPK: mitogen-activated protein kinase; EGR-1: Early growth response protein 1; NaPi-2a: Sodium-dependent phosphate transport protein 2a; 1α(OH)ase: 1α-hydroxylase; FGF23: fibroblast growth factor 23.

**Figure 2 nutrients-14-04477-f002:**
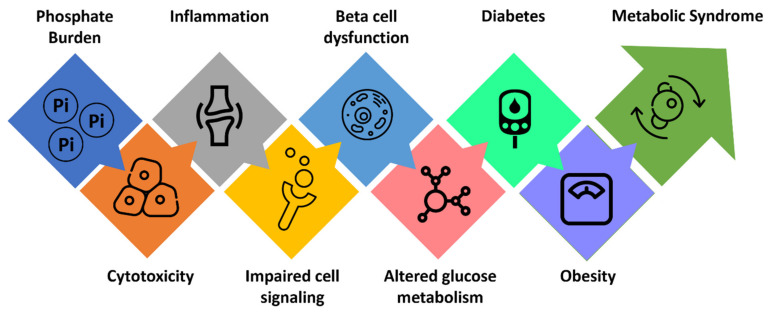
Phosphate burden-generated events which potentiate clusters of conditions commonly encountered in metabolic syndrome. Please note that some of the events shown in the figure might occur simultaneously.

## Data Availability

Not applicable.

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
