# Peer review of "Phosphate Dysregulation and Metabolic Syndrome"

_nutrients, 2022, doi:10.3390/nu14214477_

Round 1
Reviewer 1 Report
The manuscript entitled "Phosphate dysregulation and metabolic syndrome" by Mironov et al et al. (nutrients-1943754) briefly aims to review the association between high phosphate consumption and the risk of metabolic syndrome. This is a topic of interest as obesity and its associated risk for several diseases is a major burden in our society. The major concern of this brief review is that the authors, as phosphate experts, focus their review on phosphate but not on the hormones that regulate phosphate homeostasis. There are small evidences of the role of PTH and FGF23 in energy metabolism which should be included in the article rather than reviewing again briefly the impact of phosphate on cardiovascular disease and mortality which is not the main topic of the review and has been reviewed elsewhere multiple times.
Major comments:
1. As said previously it is desirable that the topic is reviewed and not other subjects such as phosphate regulation which is a nice introduction of the role of PTH, FGF23 and calcitriol that are not further reviewed or discussed. Also the references to phosphate intake and cardiovascular risk and mortality have been elsewhere reviewed multiple times and are not the focus of this brief review.
2. Revise lines 36-40, first the study mentioned in line 40 gives only in vitro evidence which should be stated. Second, the statement made is too strong for the current evidence available.
3. Reference 9 shows that the relationship between phosphate and the risk of developing diabetes type 2 was marginally significant. Moreover, only elevated calcium concentration in serum and calcium-phosphate product were associated with developing diabetes type 2 but not phosphate. Therefore, the statement in lines 51-53 is incorrect as if there is an association this seem to be very weak as seen in this study.
4. Lines 105-109. As this is one of the key questions of this review detailed evidence should be given and not a citation to another brief review from the same authors. Please expand.
Minor comments:
1. In order to evaluate the statements, mention along the manuscript if the evidence was gained from in vivo, ex vivo or in vitro studies. This is not always done
2. Almost 25 % are self-citations. Please reconsider the references given.
3. Line 86, reference 5 is not the adequate manuscript to cite with this statement.
Author Response
The manuscript entitled "Phosphate dysregulation and metabolic syndrome" by Mironov et al et al. (nutrients-1943754) briefly aims to review the association between high phosphate consumption and the risk of metabolic syndrome. This is a topic of interest as obesity and its associated risk for several diseases is a major burden in our society. The major concern of this brief review is that the authors, as phosphate experts, focus their review on phosphate but not on the hormones that regulate phosphate homeostasis. There are small evidences of the role of PTH and FGF23 in energy metabolism which should be included in the article rather than reviewing again briefly the impact of phosphate on cardiovascular disease and mortality which is not the main topic of the review and has been reviewed elsewhere multiple times.
Major comments:
- As said previously it is desirable that the topic is reviewed and not other subjects such as phosphate regulation which is a nice introduction of the role of PTH, FGF23 and calcitriol that are not further reviewed or discussed. Also the references to phosphate intake and cardiovascular risk and mortality have been elsewhere reviewed multiple times and are not the focus of this brief review.
=> We want to thank the reviewer for suggesting this important point. We’ve briefly addressed this issue in our revised manuscript.
- Revise lines 36-40, first the study mentioned in line 40 gives only in vitro evidence which should be stated. Second, the statement made is too strong for the current evidence available.
=> Modified as recommended.
- Reference 9 shows that the relationship between phosphate and the risk of developing diabetes type 2 was marginally significant. Moreover, only elevated calcium concentration in serum and calcium-phosphate product were associated with developing diabetes type 2 but not phosphate. Therefore, the statement in lines 51-53 is incorrect as if there is an association this seem to be very weak as seen in this study.
=> Modified as recommended.
- Lines 105-109. As this is one of the key questions of this review detailed evidence should be given and not a citation to another brief review from the same authors. Please expand.
=> Modified as recommended.
Minor comments:
- In order to evaluate the statements, mention along the manuscript if the evidence was gained from in vivo, ex vivo or in vitro studies. This is not always done.
=> Thank you for pointing out, we’ve revised the text accordingly.
- Almost 25 % are self-citations. Please reconsider the references given.
=> Although we believe we only cited relevant references, as pointed out, we modified the reference section.
- Line 86, reference 5 is not the adequate manuscript to cite with this statement.
=> Modified.
Reviewer 2 Report
The manuscript “Phosphate dysregulation and metabolic syndrome” describes an interesting problem of the interconnections between the phosphate levels and metabolic syndrome. The work is concise but enlightens most important aspects of the topic at a sufficient level.
I recommend publishing after considering some minor remarks.
Minor issues:
-
Could you explain the sentence? “Mineral ion deficiency in clinical medicine, including Ca2+ , Mg2+ , zinc (Zn2+), and iron (Fe3+), is common in healthy individuals (56-61).”
-
The authors could consider to extend the analyzed literature, e.g.
Osadnik, K., Osadnik, T., Delijewski, M., Lejawa, M., Fronczek, M., Reguła, R., ... & Pawlas, N. (2020). Calcium and phosphate levels are among other factors associated with metabolic syndrome in patients with normal weight. Diabetes, Metabolic Syndrome and Obesity: Targets and Therapy, 13, 1281.
Yoo, K. D., Kang, S., Choi, Y., Yang, S. H., Heo, N. J., Chin, H. J., ... & Lee, H. (2016). Sex, age, and the association of serum phosphorus with all-cause mortality in adults with normal kidney function. American journal of kidney diseases, 67(1), 79-88.
Håglin, L., Törnkvist, B., & Bäckman, L. (2020). Obesity, smoking habits, and serum phosphate levels predicts mortality after life-style intervention. Plos one, 15(1), e0227692.
Cheungpasitporn, W., Thongprayoon, C., Mao, M. A., Kittanamongkolchai, W., Sakhuja, A., & Erickson, S. B. (2018). Admission serum phosphate levels predict hospital mortality. Hospital Practice, 46(3), 121-127.
Author Response
The manuscript “Phosphate dysregulation and metabolic syndrome” describes an interesting problem of the interconnections between the phosphate levels and metabolic syndrome. The work is concise but enlightens most important aspects of the topic at a sufficient level.
I recommend publishing after considering some minor remarks.
=> Thank you for finding our manuscript meaningful.
Minor issues:
Could you explain the sentence? “Mineral ion deficiency in clinical medicine, including Ca2+ , Mg2+ , zinc (Zn2+), and iron (Fe3+), is common in healthy individuals (56-61).”
=> The text is modified in the revised submitted version.
The authors could consider to extend the analyzed literature, e.g.
Osadnik, K., Osadnik, T., Delijewski, M., Lejawa, M., Fronczek, M., Reguła, R., ... & Pawlas, N. (2020). Calcium and phosphate levels are among other factors associated with metabolic syndrome in patients with normal weight. Diabetes, Metabolic Syndrome and Obesity: Targets and Therapy, 13, 1281.
Yoo, K. D., Kang, S., Choi, Y., Yang, S. H., Heo, N. J., Chin, H. J., ... & Lee, H. (2016). Sex, age, and the association of serum phosphorus with all-cause mortality in adults with normal kidney function. American journal of kidney diseases, 67(1), 79-88.
Håglin, L., Törnkvist, B., & Bäckman, L. (2020). Obesity, smoking habits, and serum phosphate levels predicts mortality after life-style intervention. Plos one, 15(1), e0227692.
Cheungpasitporn, W., Thongprayoon, C., Mao, M. A., Kittanamongkolchai, W., Sakhuja, A., & Erickson, S. B. (2018). Admission serum phosphate levels predict hospital mortality. Hospital Practice, 46(3), 121-127.
=> As correctly pointed out we’ve included references in our revised manuscript, with relevant explanations. Thank you for the suggestion.